# HDEE: Heterogeneous Domain Expert Ensemble

**Oğuzhan Ersoy, Jari Kolehmainen & Gabriel Passamani Andrade**
Gensyn
{oguzhan,jari,gabriel}@gensyn.ai

## Abstract

Training dense LLMs requires enormous amounts of data and centralized compute, which introduces fundamental bottlenecks and ever-growing costs for large models. Several studies aim to reduce this dependency on centralization by reducing the communication overhead of training dense models. Taking this idea of reducing communication overhead to a natural extreme, by training embarrassingly parallelizable ensembles of small independent experts, has been shown to outperform large dense models trained in traditional centralized settings. However, existing studies do not take into account underlying differences amongst data domains and treat them as monolithic, regardless of their underlying complexity, size, or distribution. In this paper, we explore the effects of introducing heterogeneity to these ensembles of domain expert models. Specifically, by allowing models within the ensemble to vary in size–as well as the number of training steps taken depending on the training data's domain–we study the effect heterogeneity has on these ensembles when evaluated against domains included in, and excluded from, the training set. We use the same compute budget to train heterogeneous ensembles and homogeneous baselines for comparison. We show that the heterogeneous ensembles achieve the lowest perplexity scores in 20 out of the 21 data domains used in the evaluation. Our code is available at https://github.com/gensyn-ai/hdee.

## 1 Introduction

Large Language Models (LLMs) have seen significant improvements in performance on both language and cognitive tasks in recent years (Radford et al., 2018; Touvron et al., 2023; Brown et al., 2020; Chowdhery et al., 2023; Devlin et al., 2019; Shoeybi et al., 2019). These performance boosts can largely be accredited to aggregating a large number of GPUs (and wall-clock time) in order to train LLMs on an ever-growing corpus of data. However, the unprecedented scale of data–and the concentration of compute power required to train these models (while running ablations, etc.)–introduces costs that are infeasible for all but a handful of companies.

In order to reduce the current dependency on centralized compute, and thereby improve cost efficiency in large model training, several works aim to reduce the communication overheads of parallelized training. Methods that cut down the synchronization frequency within data parallel training have shown significant promise and have successfully trained large models using highly distributed resources (Douillard et al., 2023; Peng et al., 2024; Jaghouar et al., 2024b). By reducing synchronization frequency, it is possible to train LLMs over nodes that are geo-distributed while achieving similar throughput compared to centralised training (Jaghouar et al., 2024a). Furthermore, techniques such as mixture of experts (MoE) and ensembling enable us to combine several models in order to facilitate efficient training and inference (Cai et al., 2024). By using these techniques, and by reducing synchronization frequency to a natural extreme, independently trained domain-specific expert models can outperform large dense models by being combined (i.e. parameter averaging or MoE) or ensembled (Li et al., 2022; Sukhbaatar et al., 2024). Despite promising results shown by domain-specific experts, the effects of heterogeneity when training the experts remains largely unexplored other than for specific architectures (Wang et al., 2024).

In this paper, we explore the effects of heterogeneity in ensembles of domain expert models. Specifically, we investigate the impact of varying model sizes and total training steps depending on the training data domain's "difficulty".[1] This paper analyses two special cases of heterogeneity that correspond to (simplifications of) real-world constraints and compares them against a homogeneous baseline, leaving scope to compare additional domain-specific heterogeneities in future work. Specifically, the three cases we compare in this paper are: (i) $M_{Ho}$-$I_{Ho}$ (baseline): homogeneous model sizes and an equal number of steps for all models, (ii) $M_{Ho}$-$I_{He}$: homogeneous model sizes and unequal number of steps, (iii) $M_{He}$-$I_{Ho}$: heterogeneous model sizes and equal number of steps.

Our results show that both forms of heterogeneity considered ($M_{He}$-$I_{Ho}$ and $M_{Ho}$-$I_{He}$) achieve the lowest perplexity compared to the homogeneous baseline ($M_{Ho}$-$I_{Ho}$) in 20 out of 21 domains. Among the heterogeneous models, $M_{Ho}$-$I_{He}$ usually outperforms the other methods, especially in the difficult domains. We find that $M_{He}$-$I_{Ho}$ performs slightly worse than $M_{Ho}$-$I_{He}$ in difficult domains, but for some evaluation-only datasets, it achieves the best perplexity results. Finally, our results show that increasing the heterogeneity level (the differences in model size or iterations) improves the performance of the ensemble.

## 2 ELMFORESTS, BTM, AND HETEROGENEITY

We explore the effects of heterogeneity in ELMForests–embarrassingly parallel ensembles of expert language models (ELMs) first introduced by Li et al. (2022). ELMForests are defined by a set $\mathcal{E} = \{\text{Expert}^i\}_{i=1}^n$ of ELMs, where each $\text{Expert}^i$ is independently trained on a specialized (sub-)domain $D_i$ of a corpus $\mathcal{D} = \{D_1, \ldots, D_n\}$. As with Gururangan et al. (2021) and Li et al. (2022) we define domains based on *data provenance* (e.g. whether a document came from a computer science publication vs. a news article), and hence our ELMs are domain experts over interpretable segments of data.

The training of ELMForests is embarrassingly parallel and incremental due to the Branch-Train-Merge (BTM) algorithm introduced by Li et al. (2022). Starting from a pre-trained seed model $\text{Expert}^0$, BTM can be intuitively summarized as a three step process:

- **Step** 1 (**Branch**): For each domain $D_i \in \mathcal{D}$ sprout a new ELM from the seed model (e.g. create a "clone" to be independently trained) or, if prior iterations of BTM have already been executed, from a function of the existing expert set $\mathcal{E}$.

- **Step** 2 (**Train**): Each $\text{Expert}^i$ is trained on data from its corresponding domain $d_i$. Critically, each $\text{Expert}^i$ is trained completely independently from any other ELM, i.e. no other ELMs (of the same branch training step) are involved in the training nor are any other data domains used.

- **Step** 3 (**Merge**): Form the ELMForest $\mathcal{E}$ by combining all of the independently trained ELMs.

In future iterations of the BTM algorithm, we can incrementally grow the ELMForest to capture new data domains by following this same process.

After training, ELMForests can perform inference in two natural ways - either by ensembling output probabilities across ELMs or aggregating all models in the ELMForest into a single LM via parameter averaging. In this paper, for simplicity in adapting to heterogeneous model settings, we focus on the former and utilize a simplified variant of the cached prior method proposed by both Gururangan et al. (2021) and Li et al. (2022). Taking a probabilistic formulation of language modelling, where we estimate $p(X_t|\mathbf{x}_{<t})$ and introduce a domain variable $D$ alongside each sequence $\mathbf{x}$, we can denote the next-step conditional distribution over the history $\mathbf{x}_{<t}$ as:

$$p(X_t|\mathbf{x}_{<t}) = \sum_{i=1}^n p(X_t|\mathbf{x}_{<t}, D = D_i) \cdot p(D = D_i|\mathbf{x}_{<t}) . \tag{1}$$

---

[1]In this paper we assign difficulty levels based on the perplexities of the models with respect to each task-specific domain and the similarities between the pre-training dataset and domain-specific dataset. For example, we consider the Mathematics dataset (from S2ORC) (Reid et al., 2022) more difficult than the Tiny Stories dataset (Eldan & Li, 2023).

The ELMForest $\mathcal{E}$ naturally defines $p(X_t|\mathbf{x}_{<t}, D = D_i)$, but the **domain posterior** $p(D = D_i|\mathbf{x}_{<t})$ needs to be estimated. To do this, we use Bayes' rule:

$$p(D = D_i|\mathbf{x}_{<t}) = \frac{p(\mathbf{x}_{<t}|D = D_i) \cdot p(D = D_i)}{p(\mathbf{x}_{<t})} = \frac{p(\mathbf{x}_{<t}|D = D_i) \cdot p(D = D_i)}{\sum_{j=1}^{n} p(\mathbf{x}_{<t}|D = D_j) \cdot p(D = D_j)} . \quad (2)$$

The likelihood over sequences given a domain label are computed using the ELMs. To compute the prior we experimented with several approaches, including the exact cached priors used by Gururangan et al. (2021) and Li et al. (2022), but ultimately found that simply using a uniform prior over domains led to the best results, i.e. $p(D = D_i) = 1/n$ for all $i \in [1, \ldots, n]$.

All results reported below are from ELMs trained using BTM and ensembled via the domain posterior (Eq. 1) with a uniform domain prior.

## 2.1 HETEROGENEOUS DOMAIN EXPERT ENSEMBLES

The ELMForests studied in Li et al. (2022) are entirely homogeneous; all ELMs in the forest are the same size and trained over the same amount of steps per domain, which we refer to as our baseline $\mathtt{M_{Ho}}\text{-}\mathtt{I_{Ho}}$. We explore two heterogeneous training settings for ELMForests, where the model sizes or the number of training steps per model may differ across domains. In $\mathtt{M_{Ho}}\text{-}\mathtt{I_{He}}$ we keep the model sizes the same, but train *easy* (resp. *difficult*) domains with fewer (resp. more) steps and thereby effectively force ELMs to utilize less (resp. more) data. In $\mathtt{M_{He}}\text{-}\mathtt{I_{Ho}}$ we keep the number of steps the same, but use different expert sizes depending on whether a data domain is *easy* (i.e. smaller models) or *difficult* (i.e. larger models).

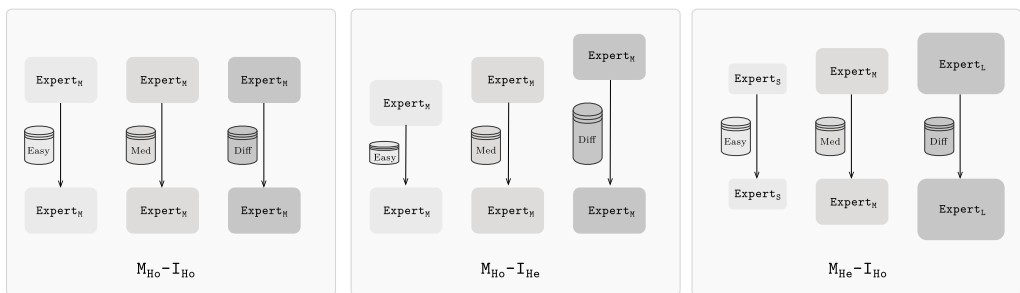

Figure 1: An iteration of BTM-style domain training in HDEE. In $\mathtt{M_{Ho}}\text{-}\mathtt{I_{Ho}}$ all models are the same size and are trained for the same number of steps. In $\mathtt{M_{Ho}}\text{-}\mathtt{I_{He}}$ all models are the same size, but are trained for more or fewer steps depending on the data domain. In $\mathtt{M_{He}}\text{-}\mathtt{I_{Ho}}$ models are different sizes depending on the data domain they will specialize in, but they are all trained for the same number of steps.

In Figure 1, we illustrate BTM-style training iterations for the three cases we consider when three model sizes are used. In theory, the number of models can be arbitrary, here we use three model sizes ($\mathtt{Expert_S}$, $\mathtt{Expert_M}$ and $\mathtt{Expert_L}$) and similarly three different numbers of training steps ($\mathtt{Iter_S}$, $\mathtt{Iter_M}$ and $\mathtt{Iter_L}$).

## 3 EXPERIMENTAL SETUP

In our experiments, we analyse whether heterogeneity in the expert forest improves the perplexity on the trained (and evaluation-only) domains. To test heterogeneity and compare with the homogeneous baseline, we assume that the trainer has a fixed compute budget for training and uses the same budget in each case. Here, considering that the computationally heavy part of a transformer layer is the FFN, we train each scenario by ensuring the following equations:

$$|\mathtt{FFN_S}| \cdot \mathtt{Iter_M} \approx |\mathtt{FFN_M}| \cdot \mathtt{Iter_S} \quad \text{and} \quad |\mathtt{FFN_L}| \cdot \mathtt{Iter_M} \approx |\mathtt{FFN_M}| \cdot \mathtt{Iter_L}$$

where $|\mathtt{FFN_i}|$ corresponds to the size of FFN layer of $\mathtt{Expert_L}$.

| S2ORC | | Wiki | | Other | |
|---|---|---|---|---|---|
| Name | Tokens | Name | Tokens | Name | Tokens |
| Mathematics | 1.4B | History & events | 226M | Caselaw | 14.5B |
| Physics | 737M | Human activities | 343M | Simple wikipedia | 70M |
| Computer science | 1.1B | Philosophy | 165M | Tiny stories | 1.3B |

Table 1: Summary of training datasets and their sizes in tokens.

In the $M_{Ho}$-$I_{Ho}$ case, we have three models of the same size $Expert_M$, each trained for $Iter_M$ steps per domain. In the $M_{Ho}$-$I_{He}$ case, we have again three models of the same size $Expert_M$ and *easy*, *moderate*, *difficult* domains are trained for $Iter_S$, $Iter_M$ and $Iter_L$ steps respectively. In the $M_{He}$-$I_{Ho}$ case, we have three models of sizes $Expert_S$, $Expert_M$ and $Expert_L$ and *easy*, *moderate*, *difficult* domains are trained for $Iter_M$ steps on their corresponding models. Finally, for the next iterations in HDEE, the experts are trained over the ones from same difficulty domains, e.g., in $M_{He}$-$I_{Ho}$ setting, a *difficult* domain expert will be trained over the latest trained $Expert_L$ of that forest.

## 3.1 DOMAINS

Table 1 shows a summary of the different datasets used in this study for domain adaptation (Reid et al., 2022). We designate the S2ORC datasets as '*difficult*, Wiki datasets as *moderate*, and the remaining datasets as *easy*. The designation is based on the seed model perplexities for the datasets that were trained using the OpenWebText corpus. For the validation and testing data, we hold out $\sim 10M$ tokens from each corpus that are not used for training.

For training, we use a selection of datasets from M2D2 (Reid et al., 2022); Caselaw (Enrico Shippole, 2024); Tiny stories (Eldan & Li, 2023); Simple wikipedia (Hwang et al., 2015).[2] In addition to the aforementioned training domains, we also evaluate the ensemble models in an out-of-domain scenario (evaluation-only) using other datasets from (Reid et al., 2022); Fineweb (Vadlapati, 2024); Hacker news (Stoddard, 2015); CC news (Zeng et al., 2024); and Reddit (Baumgartner et al., 2020).

## 3.2 MODELS AND ITERATIONS

**Seed Models** All seed models use the Llama architecture and are pre-trained with the OpenWeb-Text corpus (Dubey et al., 2024; Gokaslan et al., 2019). They share the same vocabulary size (128000) and sequence length (1024). Input text is tokenized using the sentence piece tokenizer from (Dubey et al., 2024). Table 2 shows the seed model hyperparameters. Pre-training of the seed models, consisting of $20,000$ optimizer steps, is performed using linear warm-up and a cosine annealing learning rate scheduler. The warm-up schedule consists of $1,000$ steps at the beginning of the training and the cosine annealing scheduler is set to reduce learning rate by one magnitude over remaining training steps. We use bfloat16 for numerical precision and flash-attention from Dao et al. (2022) for attention head computations.

**Trained Domain Expert Models** After pre-training, the seed models are trained using the datasets listed in Table 1. Each model is trained with a single domain and the data is not mixed within any single training run. The training batch size is kept the same as listed in Table 2, but the maximum learning rate is reduced to the final learning rate of the pre-training. We also employ warm-up and cosine annealing learning rate schedulers similar to the seed pre-training, but reduce the number of warm-up steps from 1000 to 50 steps and the total number of optimizer steps from $20,000$ to 600. For the second and third iterations of the domain training, each expert is trained over the previous iteration of the same category of expert. To avoid overfitting on the previous domains, we use the checkpoints of the previous experts at 400 steps, which is also the commonly used checkpoint for evaluating the experts (depending on the case, experts at different steps are evaluated in experiments, disclosed in results).

---

[2]Simple Wikipedia may have overlapping data with the domains from Wikipedia. Considering the same applies to all ensemble cases, and also considering the significant difference in perplexities of these datasets, this should not have any meaningful impact in the results.

| Model | 5M | 7.5M | 10M | 12.5M | 15M | 90M | 115M | 135M |
|---|---|---|---|---|---|---|---|---|
| Hidden size | 272 | 272 | 320 | 330 | 340 | 768 | 768 | 768 |
| Intermediate size | 1088 | 1088 | 1280 | 1320 | 1360 | 2304 | 3072 | 3840 |
| Attention heads | 8 | 8 | 10 | 11 | 10 | 12 | 12 | 12 |
| Number of layers | 4 | 6 | 6 | 7 | 8 | 12 | 12 | 12 |
| Batch size | 262k | 262k | 262k | 262k | 262k | 688k | 688k | 688k |
| (Max) Learning rate | .005 | .005 | .005 | .005 | .005 | .0006 | .0006 | .0006 |

Table 2: Seed model hyper-parameters used for training. Model size denotes the number of transformer layer parameters, excluding both input and output embeddings. Batch size is expressed as the number of tokens in one optimizer step.

| | Dataset | Tiny (Spread) | | | Tiny (Close) | | | Small (Close) | | |
|---|---|---|---|---|---|---|---|---|---|---|
| | | $M_{He}$-$I_{Ho}$ | $M_{Ho}$-$I_{Ho}$ | $M_{Ho}$-$I_{He}$ | $M_{He}$-$I_{Ho}$ | $M_{Ho}$-$I_{Ho}$ | $M_{Ho}$-$I_{He}$ | $M_{He}$-$I_{Ho}$ | $M_{Ho}$-$I_{Ho}$ | $M_{Ho}$-$I_{He}$ |
| Trained Domains | Math | **46.0** | 46.2 | **46.0** | 46.7 | 46.2 | **45.8** | 32.0 | 31.9 | **31.5** |
| | Physics | **72.2** | 73.2 | 72.6 | 73.3 | 73.2 | **72.6** | **48.0** | 48.1 | **48.0** |
| | CS | 61.0 | 61.6 | **60.8** | 61.6 | 61.6 | **61.3** | 40.6 | 40.5 | **40.4** |
| | History | 68.8 | 68.8 | 68.8 | 68.8 | 68.8 | 68.8 | 41.9 | 41.9 | 41.9 |
| | Human Activities | 66.7 | 66.7 | 66.7 | 66.7 | 66.7 | 66.7 | 40.3 | 40.3 | 40.3 |
| | Philosophy | 64.6 | 64.6 | 64.6 | 64.6 | 64.6 | 64.6 | 39.2 | 39.2 | 39.2 |
| | Caselaw | 74.2 | 67.7 | **64.7** | 72.4 | 67.7 | **67.1** | 33.1 | 31.8 | **31.6** |
| | Simple Wiki. | 49.8 | 47.5 | **45.6** | 48.8 | 47.5 | **47.0** | 24.0 | 23.1 | **23.0** |
| | TinyStories | 9.3 | **8.8** | 9.6 | 9.2 | **8.8** | 9.0 | 6.2 | **6.1** | 6.3 |
| Evaluation-Only Domains | Quant. Bio | 73.7 | 74.2 | **73.1** | 74.2 | 74.2 | **73.7** | 46.6 | 46.7 | **46.5** |
| | Astro-Physics | 73.2 | 74.1 | **73.1** | 73.7 | 74.1 | **73.4** | **46.7** | 47.0 | 47.0 |
| | Cond. Matter | **53.1** | 53.7 | 53.3 | 53.8 | 53.7 | **53.1** | 37.3 | 37.3 | **37.1** |
| | Statistics | 42.8 | 43.2 | **42.7** | 43.1 | 43.2 | **42.8** | 37.9 | 30.1 | **29.7** |
| | Natural Sciences | 85.3 | 85.3 | 85.3 | 85.3 | 85.3 | 85.3 | 49.7 | 49.7 | 49.7 |
| | Tech. & Appl. Sci. | 70.6 | 70.6 | 70.6 | 70.6 | 70.6 | 70.6 | 42.5 | 42.5 | 42.5 |
| | Social Sciences | 62.8 | 62.8 | 62.8 | 62.8 | 62.8 | 62.8 | 38.1 | 38.1 | 38.1 |
| | Culture Arts | 63.3 | 63.3 | 63.3 | 63.3 | 63.3 | 63.3 | 37.2 | 37.2 | 37.2 |
| | Fineweb | 150.3 | 144.1 | **139.9** | 147.7 | 144.1 | **143.2** | 61.2 | 59.5 | **59.2** |
| | Hacker News | **115.6** | 118.8 | 118.2 | **110.1** | 118.8 | 118.5 | **28.7** | 29.3 | 29.3 |
| | CC news | 107.4 | 104.2 | **102.0** | 104.4 | 104.2 | **103.8** | 41.4 | 40.1 | **39.8** |
| | Reddit | 134.9 | 134.6 | **134.5** | 134.8 | **134.6** | 134.6 | 73.4 | 73.4 | 73.4 |

Table 3: Perplexity results of the ensemble models for the evaluated data domains.

## 4 RESULTS

We test all three expert forests via three setups:

- **Tiny Spread** where the model sizes are $\text{Expert}_S$: 5M, $\text{Expert}_M$:10M and $\text{Expert}_L$:15M and training steps are $\text{Iter}_S$:200, $\text{Iter}_M$:400 and $\text{Iter}_L$:600.

- **Tiny Close** where the model sizes are $\text{Expert}_S$: 7.5M, $\text{Expert}_M$:10M and $\text{Expert}_L$:12.5M and training steps are $\text{Iter}_S$:300, $\text{Iter}_M$:400 and $\text{Iter}_L$:500.

- **Small Close** where the model sizes are $\text{Expert}_S$: 90M, $\text{Expert}_M$:115M and $\text{Expert}_L$:135M and training steps are $\text{Iter}_S$:300, $\text{Iter}_M$:400 and $\text{Iter}_L$:500.

For each setup, we train three iterations of the corresponding forest (starting with seed models). In the $i^{th}$ iteration, we use the domains given in the $i^{th}$ row of Table 1. The perplexity results after all three iterations are given in Table 3. Note that these perplexities are based on the ensemble of the final experts (each trained on three domains) using the domain posterior (Eq. 1) with a uniform domain prior. Our code is available at https://github.com/gensyn-ai/hdee.

The experimental results show that the heterogeneous approaches ($\mathtt{M_{He}}$-$\mathtt{I_{Ho}}$ and $\mathtt{M_{Ho}}$-$\mathtt{I_{He}}$) achieve the best results in almost all of the trained domains (8 out of 9) and in all 12 evaluation-only domains. The baseline $\mathtt{M_{Ho}}$-$\mathtt{I_{Ho}}$ performs better only for the Tiny stories dataset. We observe that for the *moderate* datasets, which are all from the Wikipedia cluster of M2D2, all three methods achieve the same perplexity results. This is because they each share the same $\mathtt{Expert_M}$ expert which is trained for the same $\mathtt{Iter_M}$ number of iterations, and because of the domain posterior ensembling, those models dominate the corresponding outputs. We observe the same effect in the evaluation-only Wikipedia domains.

For the domains that are trained with larger experts or more iterations (Math, Physics and CS), $\mathtt{M_{Ho}}$-$\mathtt{I_{He}}$ achieves the best results in most cases, while for some cases $\mathtt{M_{He}}$-$\mathtt{I_{Ho}}$ has the same or slightly better perplexities. These results show that heterogeneity of the expert sizes or iterations are beneficial for such domains, which is not surprising since they undergo *more* training. Yet, for the domains that are trained *less*, $\mathtt{M_{Ho}}$-$\mathtt{I_{He}}$ achieves better results than $\mathtt{M_{Ho}}$-$\mathtt{I_{Ho}}$ for two datasets. We believe that this is caused by these expert models forgetting the previous datasets, which is also supported by the fact that $\mathtt{M_{Ho}}$-$\mathtt{I_{Ho}}$ has the lowest perplexities for the latest trained dataset.

**Degree of Heterogeneity**  The setup for Tiny Spread (5M, 10M, 15M) and Tiny Close (7.5M, 10M, 12.5M) are similar in the order of model sizes and differ at the degree of the heterogeneity. We observe that $\mathtt{M_{He}}$-$\mathtt{I_{Ho}}$ performs better in the higher degree of heterogeneity, which is most likely caused by having the largest model (15M) among Tiny setups. Whereas, $\mathtt{M_{Ho}}$-$\mathtt{I_{He}}$ performs relatively better in a low degree of heterogeneity group (18 out of 21). This implies that when the model sizes are closer, training with more data is more impactful than training with a larger model. Nonetheless, compared to the baseline $\mathtt{M_{Ho}}$-$\mathtt{I_{Ho}}$, **heterogeneous models, when combined, perform better as the degree of heterogeneity increases**.

**Expert Size**  Tiny Close (7.5M, 10M, 12.5M) and Small Close (90M, 115M, 135M) share a similar degree of heterogeneity but the order of magnitude of the model sizes is different. In general, we observe no significant difference when the expert sizes are increased by more than 10 times. Yet, $\mathtt{M_{He}}$-$\mathtt{I_{Ho}}$ performs relatively better for the evaluation-only domains when the experts are larger. This can be caused by the fact that it has the largest model (135M) and it does not overfit into the trained domains.

## 5  CONCLUSION

In this paper, we explored the effects of heterogeneity in ensembles of domain expert models regarding both the model sizes and the number of training steps. We tested heterogeneity in both model size ($\mathtt{M_{He}}$-$\mathtt{I_{Ho}}$) and the number of training iterations ($\mathtt{M_{Ho}}$-$\mathtt{I_{He}}$). Our results show that heterogeneity ($\mathtt{M_{He}}$-$\mathtt{I_{Ho}}$ and $\mathtt{M_{Ho}}$-$\mathtt{I_{He}}$) almost always (20 out of 21 domains) achieves the lowest perplexities compared with the homogeneous baseline ($\mathtt{M_{Ho}}$-$\mathtt{I_{Ho}}$).

We believe that independent and parallel training of expert models together with heterogeneity would increase the diversity of expert contributions for training state-of-the-art (ensemble) models. In this way, parties with different capacities can collectively contribute resources towards the training (and inference) of such models.

In future work, we plan to explore additional and domain-specific heterogeneities, in addition to the categorised difficulty groups evaluated in this work. Also, we intend to mix the experts in HDEE into a single MoE model to reduce the inference cost. In heterogeneous expert models, this can be achieved by keeping all the parameters the same except for the intermediate size. However, the effects of such restrictions on the model performance need further exploration.

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
