# OpenReview forum: "HDEE: Heterogeneous Domain Expert Ensemble"
_ICLR.cc/2025/Workshop/MCDC — MCDC @ ICLR 2025_

### Official Review · Reviewer_Jueg · 2025-02-26

**Rating:** 6
**Confidence:** 3
**Fit:** 5

**Summary:**

The paper demonstrates that heterogeneous ensembles achieve lower perplexity across domains versus homogeneous ensembles. Two kinds of heterogeneity are considered: expert model size and expert training steps. They are compared to homogeneous ensemble training. Each ensemble training method given a fixed compute budget. The paper explores the idea of distributing a fixed compute budget non-uniformly across a set of experts according to the difficulty of the domain.

**Reason For Giving A Higher Score:**

Simple, clear idea. Experiments show promise. This would enable more efficient allocation of compute for training of embarrassingly parallelizable ensembles of small independent experts.

**Reason For Giving A Lower Score:**

More ablations needed. Not a clear cause and effect here of greater difficulty requiring more compute as the paper claims. For example, it could simply be that non-uniform allocation of compute between experts has benefits in itself.

**Strengths And Weaknesses:**

The strongest aspect is the simplicity and clarity of the idea: take an existing idea of ELMForests and consider what happens when we vary the model size or training time of experts depending on their difficulty.
The weakest aspect is the methodology: difficulty is determined based on perplexity, which is also the thing that is being measured. Therefore, this may simply be a tautology: it may be a simple observation about the shape of the perplexity curve for different domains.

**Suggestions:**

I would like to have seen more experiments run to confirm the cause and effect of allocating more compute to more difficult domains. It would have been great to consider some ablations. For example, what happens if more compute is allocated to easier domains and less to more difficult domains? Do we get worse performance than the homogeneous baseline? Is it better? Why? What happens if they are randomly assigned different amounts of compute?

---

### Official Review · Reviewer_sVzH · 2025-02-27

**Rating:** 4
**Confidence:** 3
**Fit:** 4

**Summary:**

The authors explore the effect of heterogeneity in model ensembles, where heterogeneity is introduced at the model level (allowing variable model size) and at the training level (allowing varying training steps) depending on the difficulty of the task. The method permits better allocation of compute depending on task difficulty for the domain experts in the ensemble, resulting in lower perplexity scores in almost all domains.

**Reason For Giving A Higher Score:**

The paper is clearly written and easy to follow

**Reason For Giving A Lower Score:**

The main issue is the novelty of the work, as I believe the main lesson of the paper - that using larger models or training them for longer on harder tasks improves performance - is already well-established.

**Strengths And Weaknesses:**

Strengths:

1. The paper is clearly written, with a well defined problem setting and logical conclusions drawn from the arguments and experiments in the paper.

Weaknesses:

1. Lack of novelty and limited contribution. That using a larger model or training for longer on a harder task will produce better results is a well-established result already. Indeed we have well-substantiated results regarding the compute optimal ratios of model size, data size, and training time from scaling laws [1,2], and so this paper seems to just be a direct application of known results.

2 (somewhat more minor). Experimental validation is limited. The paper uses quite small models, starting at 5M and going up to 135M params. I appreciate the problem setting selected by the authors requires a lot of training, and so very large models may be overly burdensome, but for an empirical paper I would've hoped for models at larger scale.

[1] Kaplan et al. Scaling Laws for Neural Language Models

[2] Hoffmann et al. Training Compute-Optimal Large Language Models

**Suggestions:**

1. The authors should consider to what extent their contribution is novel. One way to build on the knowledge on compute optimal model size-training-data size could be to more closely consider how ensembling interacts with these properties. For example, are there different ensembling strategies that better leverage heterogeneity?

2. For an empirical paper, I would encourage the authors to be as thorough as possible with their experimental validation and to try avoid using very small models.

---

### Official Review · Reviewer_HLoU · 2025-02-27

**Rating:** 6
**Confidence:** 3
**Fit:** 5

**Summary:**

The paper explores the impact of heterogeneous experts (size and # of training steps) for branch-train-merge type of approaches. I think the paper is perfect fit for the workshop.

**Reason For Giving A Higher Score:**

n/a

**Reason For Giving A Lower Score:**

- The experimental setting might be explained more clearly, had to go back and forth few times and not sure to still have gotten everything right.
- All models are pretty small, experiments are still toyish.

**Strengths And Weaknesses:**

\+ perfect topic match for the workshop

\+ nicely written until experimental setting

\- experimental setting a bit dense and confusing

\- results are hard to read and might not be very surprising

**Suggestions:**

Main suggestion is to make the experimental part is a bit easier to read, I found it dense and a bit confusing.

- It took me a while to understand what were the three "setups" (Tiny spread, ...) and why they were called as such, it's not written in the paper.
- it took me another while to figure out what Tiny spread - M_he I_ho means, given that both I and M vary in the the three settings.
- Degree of heterogeneity might be better defined: is there a particular formula you are looking at?
- Some sentences are hard to grasp, e.g. "heterogeneous models, when combined, perform better as the degree of heterogeneity increases", or a bit trivial "This implies that when the model sizes are closer, training with more data is more impactful than training with a larger model."
- Hard to read results in Table 3, could you maybe compute an average?

Another suggestion is whether you really need continual training of the experts on the three datasets for each level of difficulty. Training on the three datasets jointly (each column) would make the writing easier and still be valid to demonstrate your results?

---

### Decision · Program_Chairs · 2025-03-06

**Decision:**

Accept

**Comment:**

This paper poposes improvement on BTM-style methods but exploring the effect of having heterogeneous expert models with different sizes and training budget. Some reviewers argued that the novelty is somewhat limited but most of the reviewers felt that the paper was clear and is a good fit for the workshop. We recommend fixing the font used in the paper as it deviated from the provided ICLR template font. Overall, we recommend accepting the paper.